# GRPO-based Cluster Decision Agent for Unknown-$K$ Multi-view Clustering

**Xuqian Xue** [1]   **Jun Zhang** [1]   **Qi Cai** [2]   **Zhizhong Huang** [1]   **Hongming Shan** [3]   **Junping Zhang** [1]

## Abstract

Existing contrastive multi-view clustering methods rely on a pre-defined cluster number, limiting their flexibility in real-world scenarios lacking prior knowledge. To address this, we propose GROK, a novel framework driven by a cluster decision agent for unknown-$K$ multi-view clustering. It pioneers the adaptation of group relative policy optimization (GRPO)—a reinforcement learning strategy for LLM reasoning—into the unsupervised domain to autonomously determine the optimal $K$. Specifically, the agent orchestrates the clustering process through three synergistic phases. First, in the state perception phase, we employ a structure-aware adaptive backbone to aggregate multi-view data, providing the agent with consistent and discriminative consensus observations. Second, in the group decision phase, we introduce an action space divide-and-conquer strategy and an adaptive reward function. Equipped with these mechanisms, the agent performs group sampling and relative advantage estimation within the discrete action space of candidate $K$ values, autonomously searching for the optimal $K$ via reward maximization. Finally, via geometric feedback, geometric clustering guidance mechanism transforms the agent's structural hypotheses into explicit differentiable constraints to reshape feature manifolds, thereby closing the perception-decision-feedback loop. Experimental results demonstrate that GROK achieves superior clustering performance in unknown-$K$ scenarios by autonomously exploring the cluster structure.

[1]Shanghai Key Laboratory of Intelligent Information Processing, College of Computer Science and Artificial Intelligence, Fudan University, China. [2]Shanghai Key Laboratory of Navigation and Location-based Services, School of Electronic Information and Electrical Engineering, Shanghai Jiao Tong University, China. [3]Institute of Science and Technology for Brain-Inspired Intelligence and Key Laboratory of Computational Neuroscience and Brain-Inspired Intelligence, Fudan University, China. Correspondence to: Junping Zhang <jpzhang@fudan.edu.cn>.

*Proceedings of the 43rd International Conference on Machine Learning*, Seoul, South Korea. PMLR 306, 2026. Copyright 2026 by the author(s).

## 1. Introduction

Multi-view clustering aims to uncover intrinsic data structures by exploiting the consistency among complementary heterogeneous views (Liu et al., 2025). With the prevalence of high-dimensional data (Balaji et al., 2021), deep multi-view clustering (DMVC) has established itself as the mainstream, driven by its powerful representation learning.

Contrastive multi-view clustering has established itself as the dominant DMVC paradigm, aiming to learn consistent representations by maximizing cross-view mutual information. Its methodology has evolved from instance- (Tian et al., 2020) and semantic-level (Li et al., 2021) alignment to multi-level feature integration (Xu et al., 2022) and cross-view fusion incorporating structural information (Yan et al., 2023). Recent advances have further prioritized robustness against view discrepancies (Xue et al., 2025a) and noise (Yang et al., 2025) as well as adaptability to class-imbalanced data (Xue et al., 2025b). However, despite these strides, these parametric approaches relying on a pre-defined $K$ remain constrained in unknown-$K$ open-world scenarios.

To alleviate the dependency on a pre-defined $K$, early works primarily relied on geometric or statistical heuristics (Rodriguez & Laio, 2014; Likas et al., 2003; He et al., 2010). However, these traditional methods are limited by the expressiveness of shallow features and lack a feedback mechanism for representation learning. With the development of deep learning, DeepDPM (Ronen et al., 2022) introduced a split-merge mechanism in the single-view domain to dynamically infer $K$ by fitting Gaussian mixture models. In the multi-view domain, scUNC (Hu et al., 2025) utilizes community detection strategies for $K$ estimation, while MMCVA (Ma et al., 2025b) employs monte carlo sampling. However, they are hindered by non-end-to-end cascade designs or high computational costs. Crucially, existing methods fail to dynamically adjust the search for $K$ based on the quality of the resulting cluster partitions. Therefore, establishing an end-to-end framework that jointly optimizes feature learning and $K$ estimation is essential for addressing the unknown-$K$ challenge in multi-view clustering.

To address the aforementioned challenges, we propose GROK, a novel framework that deeply couples deep multi-view representation learning with autonomous cluster number estimation. Specifically, we pioneer the introduction

of GRPO (Shao et al., 2024) into multi-view clustering to predict the optimal cluster number $K$, constructing a *Perception-Decision-Feedback* closed-loop system driven by a cluster decision agent. In the state perception phase, the agent aggregates structure-aware consensus representations to form discriminative observations. Subsequently, in the group decision phase, it utilizes single or multiple agent to execute an adaptive divide-and-conquer strategy, performing group sampling and relative advantage estimation to efficiently pinpoint the optimal cluster number $K$. Finally, via geometric feedback, the system translates the agent's decision outcomes into explicit geometric constraints to optimize representation learning. This reciprocal interaction ensures that representations evolve in a clustering-friendly direction while driving the estimated $K$ to converge to its optimal value.

**Contributions.** The main contributions of this paper are summarized as follows:

- **First GRPO-based autonomous multi-view clustering framework.** We propose GRO*K*, the first framework driven by a cluster decision agent that pioneers the integration of GRPO into the unsupervised domain.

- **Action space divide-and-conquer strategy and adaptive reward function.** To adapt to clustering tasks involving varying cluster cardinalities, we design an action space divide-and-conquer strategy coupled with an adaptive reward function.

- **Geometric clustering guidance mechanism.** To mitigate the non-stationary representation issue in RL, we introduce a geometric clustering guidance mechanism. By translating the agent's structural hypotheses into explicit differentiable constraints, this mechanism reshapes the latent manifold to be clustering-friendly.

## 2. Related Work

### 2.1. Deep Multi-view Clustering

Existing DMVC methods broadly fall into reconstruction-based and contrastive paradigms. The former recovers latent structures via deep subspace clustering (Ji et al., 2017; Abavisani & Patel, 2018; Feng et al., 2025) or probabilistic generation (Xu et al., 2021; Palumbo et al., 2024). As the dominant paradigm, contrastive multi-view clustering has evolved across three key dimensions: (1) Alignment: CMC (Tian et al., 2020), CC (Li et al., 2021), and MFLVC (Xu et al., 2022) achieve instance-, semantic-, and multi-level alignment, respectively; (2) Global aggregation: GCFAggMVC (Yan et al., 2023) and CSOT (Zhang et al., 2024) utilize self-attention (Vaswani et al., 2017) and optimal transport (Villani et al., 2008) to transcend local receptive fields; (3) Robust adaptation: SEM (Xu et al., 2023)

and AdaM (Xue et al., 2025a) handle view discrepancies, while ROLL (Sun et al., 2025), PMIMC (Yuan et al., 2025), and PROTOCOL (Xue et al., 2025b) address noisy correspondences, missing views, and class imbalanced distributions. However, these methods rely on a prior $K$, a dependency manifested as either explicit structural binding (e.g., fixed prediction heads (Xue et al., 2025b)) or implicit inference mechanisms (e.g., pseudo-label generation (Sun et al., 2025)). Such constraints render current approaches ineffective in unknown-$K$ scenarios. To address this, we propose a novel non-parametric multi-view clustering framework.

### 2.2. Cluster Number Estimation and RL

The estimation of the cluster number $K$ has evolved from traditional statistical heuristics to deep adaptive approaches (Leiber et al., 2024). Early works primarily relied on geometric or statistical rules. For example, global $k$-means (Likas et al., 2003) determines cluster centers through incremental global search, while DPC (Rodriguez & Laio, 2014) identifies density peaks via decision graphs. However, these methods are constrained by shallow feature bottlenecks and suffer from high computational burdens when processing large-scale, high-dimensional data. With the development of deep learning, in the realm of single-view clustering, DED (Wang et al., 2018) attempts to combine autoencoders with t-SNE to enhance feature representation, employing density peak algorithms to estimate $K$ within the embedding space. Subsequently, DeepDPM (Ronen et al., 2022) introduced a split-merge mechanism to dynamically infer $K$ by fitting Gaussian mixture models. In the multi-view domain, DeepEDD (Ma et al., 2025a) and ICGR (Dai et al., 2024) leverage density estimation or inter-cluster connectivity to parse structures from graph topologies. Similarly, scUNC (Hu et al., 2025) and MMCVA (Ma et al., 2025b) employ community detection strategies and monte carlo sampling for $K$ estimation, respectively. Despite these advancements, existing methods remain constrained by non-end-to-end cascade designs, dependency on specific graph structures, or excessive computational costs, failing to achieve truly efficient and adaptive structure discovery.

Reinforcement Learning (RL) provides a robust mathematical framework for addressing non-differentiable optimization problems within discrete search spaces (Sutton et al., 1998). Fundamentally, an agent interacts with the environment by executing an *action* based on the current *state*, optimizing its *policy* according to the feedback *Reward* to maximize the expected cumulative return. Following this paradigm, RGC (Liu et al., 2023) pioneered the integration of multi-armed bandit mechanisms into graph clustering, utilizing a Q-Net to greedily search for the optimal $K$. Subsequently, ICGR (Dai et al., 2024) extended this framework to multi-view graph clustering. However, these methods are

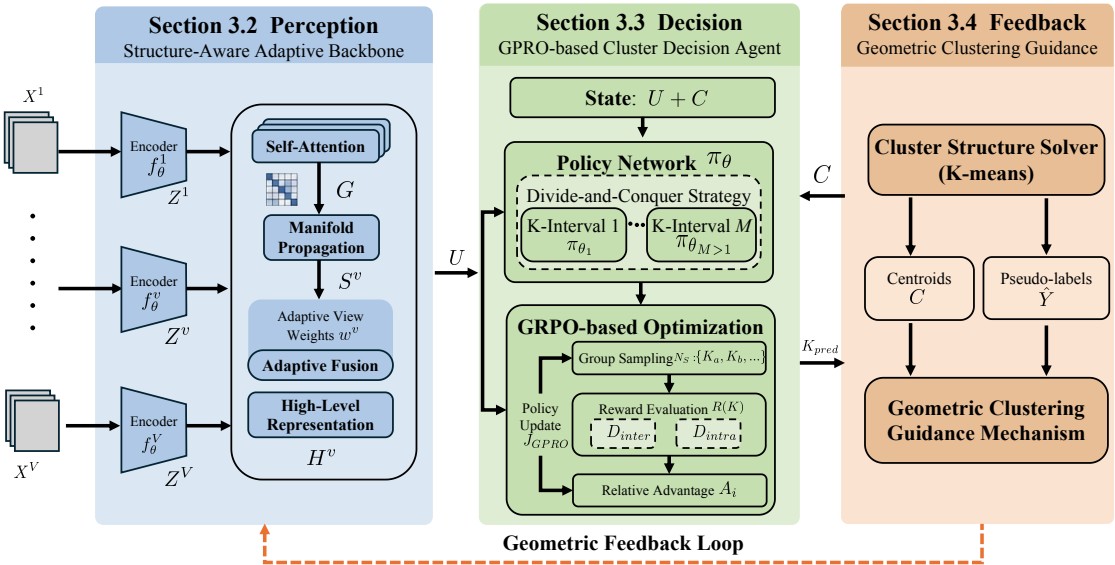

*Figure 1.* The framework of GROK.

intrinsically tailored for graph data, limiting their generalizability to generic multi-view clustering. In this work, we introduce the GRPO into the clustering domain for the first time, constructing a lightweight framework to achieve the autonomous cluster discovery for universal multi-view data.

## 3. Method

This paper proposes GROK to address the challenge of unknown cluster number $K$ in open scenarios. The framework constructs a *perception-decision-feedback* closed-loop system, as illustrated in Figure 1, comprising three core modules: (1) Structure-aware adaptive backbone: this module extracts discriminative consensus representations to provide high-quality state observations for the cluster decision agent; (2) GRPO-based cluster decision agent: this module employs GRPO to autonomously pinpoint the optimal cluster number $K$; (3) Geometric clustering guidance mechanism: this mechanism transforms the agent's structural hypotheses into explicit differentiable constraints to actively reshape the feature manifold. In the following sections, we elaborate on the specific design of each module.

### 3.1. Motivation

Existing methods typically rely on absolute metrics to determine $K$, which are highly susceptible to representation noise and local optima in open-world scenarios. To address this, inspired by the human cognitive mechanism of multi-hypothesis trial-and-comparison, we introduce GRPO (Shao et al., 2024) to reformulate structural search as a relative sequential decision process. Our core insight is that while absolute scores fluctuate due to feature non-stationarity, the relative rankings among grouped hypotheses accurately reflect the underlying structure. By sampling a group of hypotheses ($\{K_1, \ldots, K_{N_S}\}$) and evaluating their relative advantages, the agent effectively filters out baseline variance, ensuring robust identification of the intrinsic data manifold even with suboptimal representations.

### 3.2. Perception: Structure-Aware Adaptive Backbone

As the perception model, this component aims to learn consensus representations that balance view consistency and discriminability. Given multi-view data $\mathcal{X} = \{X^v\}_{v=1}^V$, we first utilize view-specific autoencoders, comprising encoders $f_{\theta^v}$ and decoders $g_{\phi^v}$, to map raw data into latent features $Z^v = f_{\theta^v}(X^v)$ and subsequently reconstruct the inputs. To capture the global structural relationships among samples, we introduce a self-attention mechanism to generate a global relation matrix $G \in \mathbb{R}^{N \times N}$. Based on this, manifold propagation is performed on $Z^v$ to yield structure-enhanced representations $S^v$. Furthermore, considering the inherent quality disparity across different views, we introduce learnable view weights $w^v$ to adaptively regulate the contribution of each view. These components are integrated into the discriminative reconstruction loss:

$$\mathcal{L}_{\mathbf{R}} = \sum_{v=1}^V \left( \|X^v - g_{\phi^v}(Z^v)\|_F^2 + w^v \|Z^v - S^v\|_F^2 \right), \quad (1)$$

where the first term preserves original information, and the second term incorporates structural constraints.

Furthermore, to obtain high-quality state observations, we first generate a consensus representation $U = \sum_{v=1}^V w^v S^v$ by aggregating structure-enhanced features across views. Simultaneously, each view feature is mapped into a high-dimensional space to obtain the projection $H^v$. On this

basis, we design a structure-guided contrastive loss:

$$\mathcal{L}_{\mathrm{C}} = -\sum_{v=1}^{V} w^v \mathbb{E}\left[ \log \frac{e^{Sm(H_i^v, U_i)/\tau}}{\sum_{j=1}^{N} (1 - G_{ij}^2) e^{Sm(H_i^v, U_j)/\tau}} \right], \quad (2)$$

where $Sm(\mathbf{a}, \mathbf{b}) = \frac{\mathbf{a}^\top \mathbf{b}}{\|\mathbf{a}\|\|\mathbf{b}\|}$ denotes the cosine similarity between two vectors. By leveraging $G_{ij}$ to prevent potential positive pairs from being wrongly repelled, this loss reshapes the feature space into highly discriminative structures, providing the cluster decision agent with high-confidence state observations.

### 3.3. Decision: GRPO-Based Cluster Decision Agent

As the decision core, the cluster decision agent is instantiated as a lightweight policy network $\pi_\theta : \mathcal{S} \to \Delta^{|\mathcal{A}|}$, responsible for inferring the optimal $K^*$ within a discrete action space $\mathcal{A} = \{2, \ldots, K_{\max}\}$. We formalize the determination of the optimal $K^*$ as a sequential decision-making process, modeled as a Markov decision process (MDP). To adapt to clustering tasks with varying scales of cluster numbers, we propose a scalable adaptive divide-and-conquer strategy. This strategy ensures robustness by dynamically configuring agent collaboration modes: the system employs single-agent inference for efficiency in low-cardinality action spaces and automatically transitions to distributed multi-agent collaboration for high-cardinality scenarios. Under this framework, the agent $\pi_\theta$ infers action probability distributions based on current states and utilizes a specialized reward function to evaluate the geometric quality of the cluster structure to guide policy updates. Next, we detail the MDP components and the GRPO-based optimization.

#### 3.3.1. State and Transition

**State** ($s_t$): The state at training epoch $t$ is defined as the fusion of multi-view consensus features and cluster centroids: $s_t = U_t + C_t$. This design enables the agent to simultaneously perceive both the feature distribution and the underlying cluster structure.

**Transition** ($\mathcal{T}$): Unlike traditional RL where the environment directly feeds back a new state, the state transition in our framework is driven by the geometric reshaping of the feature manifold. Once the agent selects an action $a_t$ (hypothesized $K$) at epoch $t$, this structural prior is fed back to the perception module via the geometric clustering guidance (Section 3.4). This forces the manifold to adapt to the new structure, yielding the subsequent state $s_{t+1}$.

#### 3.3.2. Divide-and-Conquer Action Space

The action space is defined as $\mathcal{A} = \{2, \ldots, K_{\max}\}$. To maintain efficiency across diverse tasks, we propose a scalable divide-and-conquer strategy that dynamically adjusts the

agent configuration based on the search space scale.

**Single-agent mode:** For low-cardinality datasets where $K_{\max} \leq 20$, a single lightweight agent $\pi_\theta$ manages the entire space $\mathcal{A} \in [2, 20]$. This mode prioritizes inference efficiency and minimizes computational overhead when the search space is inherently manageable.

**Multi-agent collaboration:** When $K_{\max} > 20$, a divide-and-conquer strategy automatically activates, partitioning $\mathcal{A}$ into overlapping equal-sized intervals of 20 to maintain sampling density and prevent exploration collapse. A low-region agent ($\pi_{\theta_1}$) captures macroscopic structures within $[2, K_{\mathrm{mid}}]$, while high-region agents ($\pi_{\theta_{m>1}}$) specialize in fine-grained verification for higher intervals. This 20-unit threshold is empirically optimized and verified against standard multi-view clustering benchmarks (typically ranging from 3 to 100 clusters).

This adaptive mechanism prevents exploration collapse in high-cardinality scenarios by ensuring sufficient sampling density across the entire search range, while avoiding redundant computation in simple tasks.

#### 3.3.3. Adaptive Reward Function

For each state $s_t$, the agent samples a group of $N_S$ cluster hypotheses $\mathcal{K} = \{K_1, \ldots, K_{N_S}\}$. $R(K_i)$ represents the reward score for the $i$-th hypothesis $K_i$ within this sampled group. To guide the agent away from trivial solutions, we design a adaptive reward function:

$$R(K) = \frac{\mathcal{D}_{\mathrm{inter}} - \mathcal{D}_{\mathrm{intra}}}{\psi(K)}, \quad (3)$$

where $\mathcal{D}_{\mathrm{inter}} = \frac{1}{K^2} \sum_{i,j} \mathcal{D}(c_i, c_j)$ denotes the inter-cluster separation, and $\mathcal{D}_{\mathrm{intra}} = \frac{1}{N} \sum_n \min_k \mathcal{D}(U_n, c_k)$ denotes intra-cluster compactness. $\mathcal{D}(\cdot, \cdot)$ denotes the squared Euclidean distance. While these terms provide a basic geometric evaluation, the reward remains sensitive to systemic numerical bias in high-cardinality scenarios. To address this, the denominator $\psi(K)$ applies dual-scale normalization: the low-cardinality-$K$ region utilizes a linear constraint ($\psi \propto K$) to maintain sensitivity, while the high-cardinality-$K$ region switches to a logarithmic scale ($\psi \propto \log K$). This transition prevents rewards from becoming too small as cluster distances naturally shrink at high $K$ values. By constraining the denominator, the mechanism maintains reward consistency across all agents, thereby stabilizing the joint optimization process.

#### 3.3.4. GRPO-based optimization

To achieve efficient policy updates, we adopt the GRPO algorithm (Shao et al., 2024). GRPO estimates the advantage function by sampling a group of outputs for the state $s_t$ and utilizing the group's statistical features as a

dynamic baseline, thereby significantly reducing the variance of gradient estimation. Specifically, for a given state $s_t$, the policy $\pi_\theta$ samples a group of cluster hypotheses $\mathcal{K} = \{K_1, \ldots, K_{N_S}\}$. The relative advantage $A_i$ for each $K_i$ is calculated based on its performance within the group:

$$A_i = \frac{R(K_i) - \text{mean}(R_1, \ldots, R_{N_S})}{\text{std}(R_1, \ldots, R_{N_S}) + \epsilon}. \tag{4}$$

Based on the group relative advantage $A_i$, GRPO updates the policy parameters $\theta$ by maximizing the clipped surrogate objective, which encourages high-advantage actions while ensuring training stability:

$$J_{\text{GRPO}} = \mathbb{E}_{s \sim \pi_{\theta_{\text{old}}}} \left[ \frac{1}{m} \sum_{i=1}^m \min\left( \rho_i A_i, \text{clip}(\rho_i, 1-\epsilon, 1+\epsilon) A_i \right) \right], \tag{5}$$

where $\rho_i = \frac{\pi_\theta(a_i|s)}{\pi_{\theta_{\text{old}}}(a_i|s)}$ is the probability ratio that functions as the importance sampling weight. The clip($\cdot$) function restricts this ratio to $[1-\epsilon, 1+\epsilon]$ with a hyperparameter $\epsilon$ (typically 0.2) to ensure training stability by limiting the update step size. By optimizing this group-averaged objective, the policy network can stably converge toward optimal $K$.

### 3.4. Feedback: Geometric Clustering Guidance

This module translates the discrete structural hypothesis $K_{\text{pred}}$ into differentiable geometric constraints. Once the cluster decision agent locks the current prediction $K_{\text{pred}}$, this module partitions the consensus features $U$ to generate the cluster centroid set $\mathcal{C} = \{c_1, \ldots, c_{K_{\text{pred}}}\}$ and pseudo-labels $\hat{Y}$, thereby transforming the agent's hypothesis into explicit structural guidance for the perception module. We propose a structural consistency loss to supervise the manifold reshaping:

$$\mathcal{L}_\text{S} = -\frac{1}{N} \sum_{i=1}^N \sum_{k=1}^{K_{\text{pred}}} \mathbb{I}(\hat{y}_i = k) \log(P(c_k|U_i)). \tag{6}$$

The assignment probability $P(c_k|U_i)$ is modeled under an isotropic Gaussian assumption, mapping the Euclidean distance from each feature to the centroids into a normalized distribution:

$$P(c_k|U_i) = \frac{\exp(-\|U_i - c_k\|^2/\tau)}{\sum_{j=1}^{K_{\text{pred}}} \exp(-\|U_i - c_j\|^2/\tau)}. \tag{7}$$

By minimizing $\mathcal{L}_\text{S}$, the feedback mechanism forces the perception module to optimize consensus representations to match the agent's predicted structure at the current iteration, effectively closing the perception-decision-feedback loop.

The total optimization objective is formalized as:

$$\mathcal{L}_\text{T} = \mathcal{L}_\text{R} + \mathcal{L}_\text{C} + \lambda \mathcal{L}_\text{S}, \tag{8}$$

where $\lambda$ balances the geometric constraint. This joint optimization closes the loop: decisions define structure, and structure reshapes features, enabling active correction of topological ambiguities. The detailed training process and the complete pseudocode of GROK are provided in Appendix.

*Table 1.* Summary of datasets.

| Datasets | N | V | K | Dims |
|---|---|---|---|---|
| Prokaryotic | 551 | 3 | 4 | 393,3,438 |
| Synthetic3d | 600 | 3 | 3 | 3,3,3 |
| Caltech101-7 | 1,474 | 6 | 7 | 48,40,254,1984,512,928 |
| Hdigit | 5,000 | 2 | 10 | 784, 256 |
| CIFAR10 | 50,000 | 3 | 10 | 512,2048,1024 |
| Caltech20 | 2,386 | 6 | 20 | 48,40,254,1984,512,928 |
| DHA | 483 | 2 | 23 | 110, 6144 |
| YouTubeFace | 101,499 | 3 | 31 | 64,512,64,647,838 |

## 4. Experiments

### 4.1. Experimental Setup

To evaluate the effectiveness of GROK, we conduct experiments on eight multi-view datasets: Prokaryotic (Brbić et al., 2016), Synthetic3d (Kumar et al., 2011), Caltech7 (Fei-Fei et al., 2004), Hdigit (Chen et al., 2022), CIFAR10 (Yan et al., 2023), CCV (Jiang et al., 2011), DHA (Lin et al., 2012), and YouTubeFace (Yan et al., 2023). These datasets exhibit significant diversity, with sample sizes ranging from 483 to 101,499 and cluster numbers spanning from 3 to 31, as shown in Table 1. We benchmark GROK against state-of-the-art clustering methods, including one non-parametric single-view clustering method, DeepDPM (Ronen et al., 2022), and seven parameterized multi-view clustering methods: MFLVC (Xu et al., 2021), GCFAggMVC (Yan et al., 2023), SEM (Xu et al., 2023), CSOT (Zhang et al., 2024), ROLL (Sun et al., 2025), PMIMC (Yuan et al., 2025), and AdaM (Xue et al., 2025a) (detailed in Section 2). Furthermore, we adopt Accuracy (ACC), Normalized Mutual Information (NMI), and Adjusted Rand Index (ARI) to measure clustering performance. Our code is available at https://github.com/Scarlett125/GROK.

### 4.2. Implementation Details And Network Architecture

The proposed framework, GROK, comprises three core components: a multi-view autoencoder backbone, a Transformer-based fusion module, and a scalable policy framework. For feature extraction, each view employs an encoder with four fully connected layers ($Input \rightarrow 500 \rightarrow 500 \rightarrow 2000 \rightarrow 512$) and a symmetric decoder to preserve view-specific information. To capture high-order cross-view consistency, the fusion module utilizes Transformer encoder layers to aggregate multi-view features into the consensus representation via a learnable weighted

*Table 2.* Comparison results on Prokaryotic, Synthetic3d, Caltech7, Hdigit, CIFAR10, CCV, DHA, and YouTubeFace datasets.

| Methods | Prokaryotic | | | Synthetic3d | | | Caltech7 | | | Hdigit | | |
|---|---|---|---|---|---|---|---|---|---|---|---|---|
| | ACC | NMI | ARI | ACC | NMI | ARI | ACC | NMI | ARI | ACC | NMI | ARI |
| DeepDPM | 0.618 | 0.335 | 0.213 | 0.650 | 0.465 | 0.422 | 0.585 | 0.465 | 0.323 | 0.397 | 0.411 | 0.252 |
| MFLVC | 0.522 | 0.260 | 0.172 | 0.750 | 0.409 | 0.383 | 0.434 | 0.515 | 0.323 | 0.791 | 0.806 | 0.742 |
| GCFAggMVC | 0.496 | 0.255 | 0.139 | 0.970 | 0.870 | 0.912 | 0.402 | 0.532 | 0.297 | 0.973 | 0.927 | 0.941 |
| SEM | 0.439 | 0.246 | 0.096 | 0.932 | 0.770 | 0.811 | 0.431 | 0.585 | 0.364 | 0.979 | 0.942 | 0.955 |
| CSOT | 0.531 | 0.241 | 0.164 | 0.967 | 0.860 | 0.903 | 0.360 | 0.530 | 0.291 | 0.938 | 0.868 | 0.869 |
| ROLL | 0.525 | 0.259 | 0.124 | 0.855 | 0.599 | 0.622 | 0.358 | 0.204 | 0.123 | 0.455 | 0.456 | 0.307 |
| PMIMC | 0.430 | 0.162 | 0.060 | 0.972 | 0.890 | 0.930 | 0.540 | 0.590 | 0.438 | 0.967 | 0.919 | 0.928 |
| AdaM | 0.525 | 0.324 | 0.201 | 0.973 | 0.885 | 0.922 | 0.335 | 0.469 | 0.214 | 0.982 | 0.950 | **0.962** |
| GRO*K* | **0.632** | **0.337** | **0.226** | **0.977** | **0.895** | **0.932** | **0.636** | **0.626** | **0.536** | **0.983** | **0.951** | **0.962** |

| Methods | CIFAR10 | | | CCV | | | DHA | | | YouTubeFace | | |
|---|---|---|---|---|---|---|---|---|---|---|---|---|
| | ACC | NMI | ARI | ACC | NMI | ARI | ACC | NMI | ARI | ACC | NMI | ARI |
| DeepDPM | 0.651 | 0.880 | 0.726 | 0.221 | 0.194 | 0.068 | 0.256 | 0.570 | 0.214 | 0.155 | 0.140 | 0.022 |
| MFLVC | 0.956 | 0.901 | 0.906 | 0.192 | 0.171 | 0.069 | 0.644 | 0.767 | 0.524 | 0.108 | 0.098 | 0.012 |
| GCFAggMVC | 0.990 | 0.974 | 0.979 | 0.338 | 0.325 | 0.169 | 0.781 | 0.819 | 0.640 | **0.316** | 0.319 | 0.090 |
| SEM | **0.993** | 0.979 | 0.984 | 0.366 | 0.334 | 0.185 | 0.741 | 0.837 | 0.640 | 0.305 | 0.319 | 0.088 |
| CSOT | 0.992 | 0.978 | 0.983 | 0.292 | 0.297 | 0.132 | 0.708 | 0.796 | 0.599 | 0.293 | 0.306 | 0.079 |
| ROLL | 0.830 | 0.775 | 0.746 | 0.138 | 0.097 | 0.022 | 0.807 | 0.833 | 0.689 | 0.079 | 0.048 | 0.005 |
| PMIMC | 0.989 | 0.974 | 0.977 | 0.122 | 0.093 | 0.013 | 0.681 | 0.786 | 0.553 | 0.150 | 0.152 | 0.023 |
| AdaM | 0.991 | 0.978 | 0.981 | 0.302 | 0.318 | 0.156 | **0.825** | 0.859 | 0.718 | 0.298 | 0.314 | 0.088 |
| GRO*K* | **0.993** | **0.981** | **0.985** | **0.369** | **0.336** | **0.186** | 0.816 | **0.867** | **0.721** | 0.308 | **0.325** | **0.098** |

*Table 3.* Comparison of the predicted cluster number ($K$) against the ground truth ($\Delta = K_{\text{pred}} - K_{\text{gt}}$).

| Dataset | $K_{\text{gt}}$ | DeepDPM | | GRO*K* | |
|---|---|---|---|---|---|
| | | $K_{\text{pred}}$ | $\Delta$ | $K_{\text{pred}}$ | $\Delta$ |
| Prokaryotic | 4 | 4 | 0 | 3 | −1 |
| Synthetic3d | 3 | 2 | −1 | 3 | 0 |
| Caltech7 | 7 | 5 | −2 | 4 | −3 |
| Hdigit | 10 | 11 | +1 | 10 | 0 |
| CIFAR10 | 10 | 18 | +8 | 10 | 0 |
| CCV | 20 | 26 | +6 | 21 | +1 |
| DHA | 23 | 8 | −15 | 22 | −1 |
| YouTubeFace | 31 | 19 | −12 | 30 | −1 |

layer. For autonomous cluster number estimation, the policy network employs a bimodal architecture that separately processes instance-level and cluster-center features through linear mapping and normalization. These features are aggregated into a global representation via weighted mean pooling, which finally outputs a probability distribution over candidate $K$ values via a softmax-activated layer.

The framework is implemented in PyTorch and optimized via the Adam optimizer on an NVIDIA A100 GPU workstation. The training follows a two-stage procedure. (i) Pre-training stage: autoencoders are trained for 200 epochs

using reconstruction loss (learning rate $3 \times 10^{-4}$, batch size 256) to initialize the feature space. (ii) Joint optimization stage: the model alternately performs feature alignment and optimal $K$ searching for 50 to 100 epochs. During this stage, the structure-guidance contrastive loss and clustering consistency loss are jointly optimized, with the trade-off hyper-parameter $\lambda$ empirically set to 0.05. To accommodate varying search space cardinalities, the policy framework dynamically instantiates agents based on the maximum cluster number ($K_{max}$). Policy parameters are updated using a clipped proximal policy optimization objective (clipping threshold $\epsilon = 0.2$) to maximize expected returns while ensuring training stability.

### 4.3. Main Results

Table 2 presents comparative results against one non-parametric and seven parameterized multi-view clustering methods, with Table 3 specifically detailing the cluster discovery performance between DeepDPM and GRO*K*. To ensure fairness, DeepDPM is provided with concatenated multi-view features as input. The results demonstrate that GRO*K* achieves substantial advantages across varying scales, outperforming DeepDPM in ACC by 32.7% on Synthetic3d (0.977 vs. 0.650) and 34.2% on CIFAR10 (0.993 vs. 0.651). This performance gap stems from the accuracy of cluster

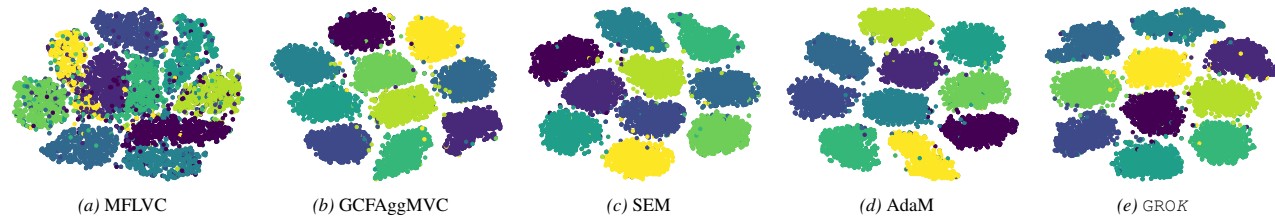

*(a) MFLVC*      *(b) GCFAggMVC*      *(c) SEM*      *(d) AdaM*      *(e) GROK*

*Figure 2.* Visualization of t-SNE on the Hdigit dataset.

*Table 4.* Ablation study of key components.

| Variants | Prokaryotic | | | Caltech7 | | | Hdigit | | | CCV | | |
|---|---|---|---|---|---|---|---|---|---|---|---|---|
| | ACC | NMI | ARI | ACC | NMI | ARI | ACC | NMI | ARI | ACC | NMI | ARI |
| Backbone (Need True $K$) | 0.524 | 0.323 | 0.201 | 0.335 | 0.469 | 0.214 | 0.982 | 0.950 | 0.962 | 0.343 | 0.352 | 0.168 |
| + Auto-K Agent | 0.537 | 0.428 | 0.315 | 0.622 | 0.620 | 0.520 | 0.900 | 0.918 | 0.901 | 0.284 | 0.325 | 0.158 |
| + Structure Guidance | **0.632** | **0.337** | **0.226** | **0.636** | **0.626** | **0.536** | **0.983** | **0.951** | **0.962** | **0.369** | **0.336** | **0.186** |

number estimation. Specifically, on the CIFAR10, while DeepDPM suffers from a significant overestimation of $+8$, our method achieves zero deviation ($\Delta = 0$). This advantage is further highlighted on the DHA dataset, where Deep-DPM's split-merge rules fail to disentangle global structures, resulting in a severe underestimation of $-15$, whereas our cluster decision agent reduces the deviation to only $-1$. This improvement validates our reinforcement learning mechanism's ability to autonomously discover $K$ values that align with the underlying distribution.

Compared to seven parameterized methods requiring a prior ground-truth $K$, GROK adaptively discovers the optimal $K$ via its cluster decision agent. Results show that GROK achieves competitive performance across most benchmarks. Notably, on class-imbalanced Prokaryotic and Caltech7, despite predicting $K = 3$ and $K = 4$ (deviating from ground truth), GROK yields significant ACC improvements of 10.1% and 9.6% over best-performing baselines. This robustness stems from our agent's ability to prioritize dominant high-density structures rather than over-fitting sparse tail noise. Admittedly, on the DHA and YouTubeFace, our ACC slightly trails the best-performing parameterized methods (e.g., 0.816 vs. 0.825 on DHA). This marginal gap is primarily caused by the slight deviation between the predicted and ground-truth cluster numbers ($\Delta = -1$ on both datasets), which imposes an inherent penalty on the ACC compared to baselines that use $K$ as a prior. Nevertheless, GROK maintains the highest NMI and ARI on both datasets. On other benchmarks, our method achieves consistent, albeit sometimes marginal, performance leads. It is worth emphasizing that our primary objective is not merely pursuing performance improvement, but endowing the model with the capability to autonomously discover cluster structures, achieving a leap from manual configuration to model autonomy while maintaining robust performance.

To intuitively assess discriminative power, we visualize the latent spaces of five methods on Hdigit via t-SNE, as shown

in Figure 2. The results demonstrate that, GROK accurately predicts the ground-truth $K$ and establishes a superior cluster structure.

### 4.4. Ablation Study

To verify the effectiveness of the core components, we compare the performance of the baseline, the model with only the cluster decision agent (+Auto-$K$), and the full model (+Structure Guidance), as shown in Table 4. The experimental results reveal two key findings: (i) Superiority of autonomous decision-making (auto-$K$): the introduction of the cluster decision agent significantly enhances performance on imbalanced datasets, with the ACC on Caltech7 jumping from 0.335 to 0.622. This strongly confirms that forcibly fitting a prior $K$ often leads to segmentation errors in tail classes, whereas the agent can autonomously discover natural structures that better align with the underlying data distribution. (ii) Necessity of structure guidance: with the incorporation of structure guidance, the performance on the CCV recovers significantly and surpasses the baseline. This confirms that the module optimizes the feature space to effectively prevent the agent from blind exploration, thereby establishing the model's robustness in complex scenarios.

### 4.5. In-depth Analysis of the Cluster Decision Agent

**Convergence analysis of cluster number estimation.** To intuitively validate the convergence and accuracy of the Cluster Decision Agent in searching for the optimal cluster number, we visualize the training dynamics of the predicted $K$ and reward changes on Hdigit, Synthetic3d and DHA datasets, as shown in Figure 3. The experimental results demonstrate that, following initial exploration, the agent rapidly converges to the ground-truth cluster number within 15 epochs via the GRPO strategy and maintains stability thereafter. Furthermore, the reward curve is highly synchronized with the convergence of $K$, confirming that the

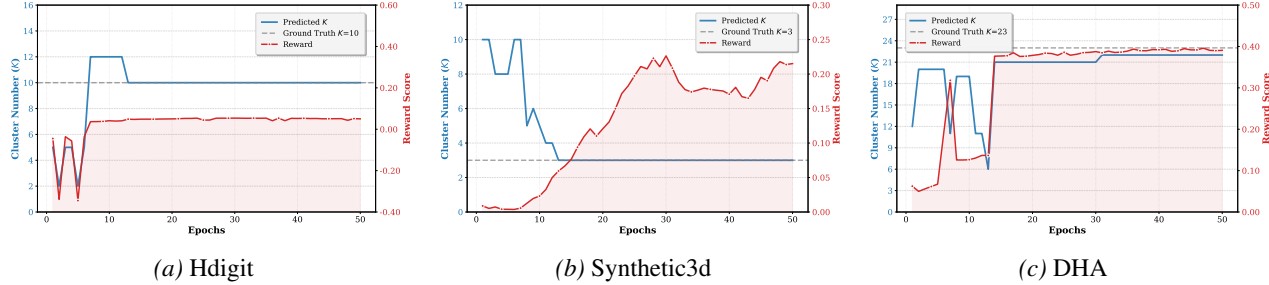

*Figure 3.* Learning curves of predicted $K$ and rewards.

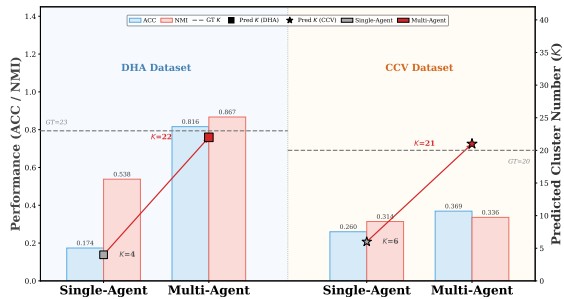

*Figure 4.* Performance comparison between single-agent and multi-agent frameworks.

geometric reward function accurately characterizes structural quality and effectively guides the model to precisely locate the optimal cluster count across different scenarios.

**Effectiveness of the divide-and-conquer strategy.** To verify the necessity of the divide-and-conquer strategy for handling tasks with a high number of clusters, we conducted comparative experiments on the DHA ($K = 23$) and CCV ($K = 20$), as shown in Figure 4. The results demonstrate that a single-agent, hindered by a low-$K$ preference, is prone to exploration collapse within vast search spaces, leading to severely underestimated $K$ values (e.g., predicting $K = 4$ for DHA). In contrast, multi-agent framework effectively decomposes the action space through a divide-and-conquer strategy, enabling the predicted values to accurately approximate the ground-truth distribution (e.g., $K = 22$ for DHA). This significantly enhances clustering robustness and performance in scenarios with large cluster numbers.

*Table 5.* Ablation study of the reward normalization mechanism.

| Variants | **Hdigit** ($K_{GT} = 10$) | | | **DHA** ($K_{GT} = 23$) | | | **CCV** ($K_{GT} = 20$) | | |
|---|---|---|---|---|---|---|---|---|---|
| | Pred $K$ | ACC | NMI | Pred $K$ | ACC | NMI | Pred $K$ | ACC | NMI |
| w/o Normalization | 19 | 0.618 | 0.841 | 39 | 0.654 | 0.825 | 38 | 0.276 | 0.336 |
| Unified Norm. | 10 | 0.981 | 0.946 | 6 | 0.261 | 0.672 | 12 | 0.318 | 0.298 |
| **Adaptive Norm. (Ours)** | **10** | **0.983** | **0.951** | **22** | **0.816** | **0.867** | **21** | **0.369** | **0.336** |

**Ablation study on adaptive reward function.** To verify the necessity of the Scale-Adaptive Reward, we conducted comparative experiments across three datasets, as shown in Table 5. The results confirm the critical role of this strategy: without normalization, the reward inflates as $K$ increases, leading to a cluster explosion (e.g., $K$ reaching 39 on DHA).

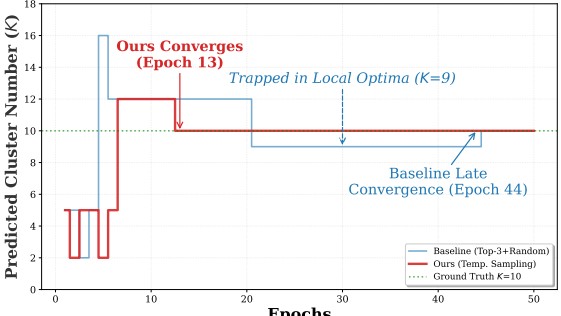

*Figure 5.* Exploration Efficiency Comparison on Hdigit.

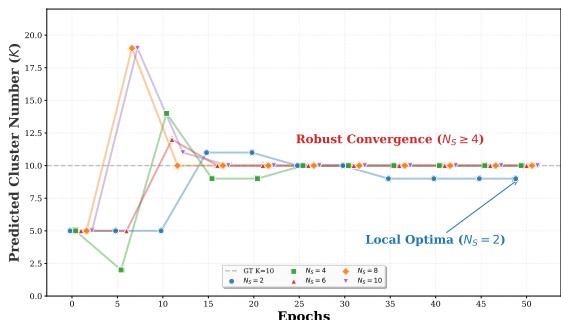

*Figure 6.* Sampling number $N_S$ analysis on Hdigit.

Conversely, employing only a unified normalization imposes excessive penalties, resulting in a low-$K$ bias (e.g., $K$ dropping to 6 on DHA). In contrast, our adaptive reward function effectively balances the search across different action spaces, enabling the agent to rapidly converge to the optimal cluster number while enhancing performance.

**Impact of sampling strategy on exploration efficiency.** To verify the efficiency and stability of the GRPO sampling mechanism, we benchmarked our temperature-based strategy against a top-k random sampling baseline on the Hdigit dataset. As shown in Figure 5, our method shortens the convergence time to the ground-truth cluster number by nearly 70% compared to the top-k baseline. Furthermore, we investigated the impact of the sampling number $N_S$ on model robustness. As illustrated in Figure 6, the agent consistently achieves stable results and successfully escapes local optima once $N_S \geq 4$. We ultimately set $N_S = 6$. These results demonstrate that temperature regulation can precisely bal-

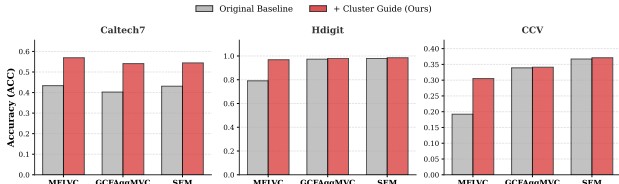

Figure 7. Comparison of the ACC of the policy model as a plug-and-play module across three baselines.

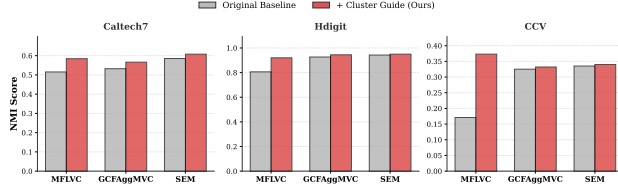

Figure 8. Comparison of the NMI of the policy model as a plug-and-play module across three baselines.

ance exploration and exploitation, significantly enhancing both the efficiency and stability of structure discovery.

### 4.6. Plug-and-Play Generalizability of the Agent

We seamlessly integrated our cluster decision agent as a plug-and-play module into three parameterized methods: MFLVC, GCFAggMVC, and SEM, as shown in Figure 7 and 8. The results demonstrate that this module enhances clustering performance, most notably achieving a 13.6% ACC improvement on Caltech7, while successfully liberating baseline methods from their rigid dependency on a prior $K$. This plug-and-play characteristic endows existing models with the capability to autonomously perceive cluster structures, marking a definitive leap from manual configuration to model autonomy.

### 4.7. Convergence and Parameter Sensitivity Analysis

To verify the stability of the training process, we visualized the evolution curves of the training loss and clustering performance (ACC, NMI) on DHA, as shown in Figure 9.

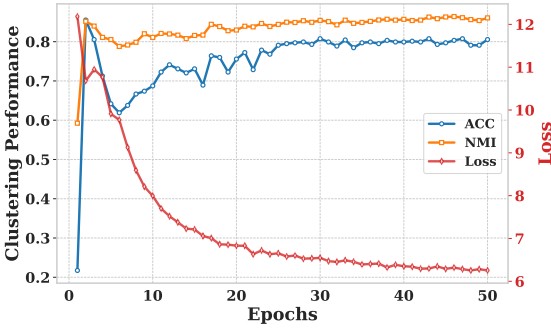

Figure 9. Convergence analysis.

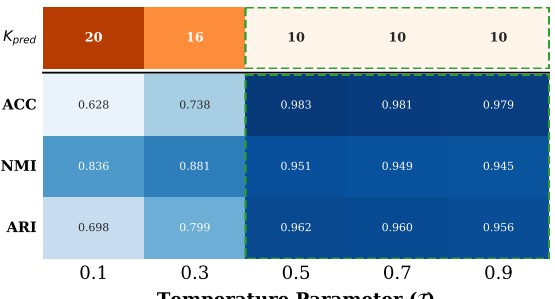

Figure 10. Parameter $\tau$ sensitivity analysis.

The results indicate that the loss function and performance metrics converge smoothly and synchronously, confirming the effectiveness of the objective function and the model's rapid, stable convergence characteristics. Furthermore, we investigated the impact of the contrastive temperature ($\tau$) on the Hdigit dataset. As illustrated in Figure 10, the model exhibits stable performance across a broad interval of contrastive temperatures $[0.5, 0.9]$, and we consequently adopted $\tau = 0.5$. Overall, GROK demonstrates robust stability and low sensitivity to hyper-parameters.

## 5. Conclusion

This paper proposes GROK, a GRPO-based cluster decision agent designed to liberate contrastive multi-view clustering from the dependency on the prior cluster number $K$. Specifically, we formulate the search for the optimal $K$ as a perception-decision-feedback co-evolutionary process. To overcome exploration collapse within discrete action spaces, GROK introduces an action space divide-and-conquer strategy coupled with an adaptive reward function, where multi-mode agents are configured to ensure comprehensive coverage across tasks with varying cluster cardinalities. Crucially, we establish a geometric feedback from the decision space to the feature manifold to actively refine feature distribution in the latent space. Extensive experiments demonstrate that GROK exhibits superior convergence efficiency and robustness. Furthermore, as a plug-and-play module, it enhances the performance of existing parameterized baselines. By endowing models with autonomous cluster decision-making capabilities, this work pioneers a novel agent-based paradigm for unsupervised representation learning.

**Acknowledgement** This work was supported by the National Natural Science Foundation of China (No. 62576103).

## Impact Statement

By effectively addressing the critical dependency on the prior ground-truth $K$ in contrastive multi-view clustering, this work enables autonomous cluster structure discovery

in complex scenarios, holding significant potential for applications in fields such as bioinformatics and multimedia analysis.

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

---

**Algorithm 1** GROK

---

**Require:** Multi-view dataset $\{X^v\}_{v=1}^{V}$, search space upper bound $K_{max}$, training epochs $T$.

**Ensure:** Optimal cluster number $K^*$, refined consensus representation $U$, and cluster labels $Y$.

 1: **Phase 1: Structure-aware Pre-training**
 2: Initialize backbone networks $\Phi$ and fusion module.
 3: **for** each pre-training epoch **do**
 4:    Extract initial consensus representation $U$ via contrastive learning (Eq. 1-2).
 5:    Update $\Phi$ to establish a discriminative feature space.
 6: **end for**
 7: **Phase 2: Joint Optimization & Fine-tuning**
 8: Initialize policy network $\pi_\theta$.
 9: **for** epoch $t = 1$ to $T$ **do**
10:    **1. State Construction (Perception):**
11:    Apply cluster structure solver (K-means) on $U$ to derive centroid set $C$.
12:    Construct state $s_t$ by mapping $U$ and $C$ into a latent geometric observation.
13:    **2. Decision Phase (GRPO Optimization):**
14:    Sample a group of candidate cluster numbers $\{K_1, K_2, ..., K_G\}$ from $\pi_\theta(s_t)$.
15:    Compute geometric rewards $\{r_1, r_2, ..., r_G\}$ using Eq. 3.
16:    Calculate relative advantages $A_i$ within the group (Eq. 4).
17:    Update policy network $\pi_\theta$ using the GRPO objective (Eq. 5).
18:    Lock the optimal prediction $K_{pred} = \arg\max \pi_\theta(s_t)$.
19:    **3. Feedback & Fine-tuning Phase:**
20:    Transform $K_{pred}$ into a differentiable geometric guidance constraint (Eq. 6).
21:    Back-propagate the guidance signal to fine-tune backbone $\Phi$.
22:    Reshape feature manifold $U$ to align with the predicted structure.
23: **end for**
24: **Return** $K^* = K_{pred}$, final representation $U$, and labels $Y$.

---

