# OpenReview forum: "GRPO-based Cluster Decision Agent for Unknown-$\boldsymbol{K}$ Multi-view Clustering"
_ICML.cc/2026/Conference — ICML 2026 regular_

### Official Review · Reviewer_3x17 · 2026-02-22

**Soundness:** 3
**Presentation:** 3
**Significance:** 3
**Originality:** 3
**Overall Recommendation:** 4
**Confidence:** 4

**Summary:**

This paper introduces GROK, a framework designed to address the critical limitation of existing multi-view clustering methods: their dependency on a pre-defined number of clusters K. In real-world open scenarios, the true K is often unknown, and current approaches either rely on rigid priors or inefficient heuristic estimations that decouple feature learning from cluster number discovery.

**Compliance With Llm Reviewing Policy:**

Affirmed.

**Final Justification:**

The authors have addressed my concerns

**Key Questions For Authors:**

1 Why choose GRPO over traditional Actor-Critic or PPO?

2 In the divide-and-conquer strategy, overlapping K intervals are adopted. How does the size of the overlapping regions affect the consistency of decision-making?

**Limitations:**

Yes. The paper should explicitly acknowledge the increased training and inference costs introduced by the RL-based agent compared to traditional heuristic K-estimation methods, and discuss scenarios where this trade-off may not be justified.

**Strengths And Weaknesses:**

Strengths:

1 unknown-K multi-view clustering is a new research problem.

2 The paper includes ablation studies that validate the necessity of each core component: the Auto-K agent.

3 The experimental design is clever and thoroughly demonstrated, effectively validating the necessity of the core modules

Weaknesses:

1 The GRPO mechanism requires sampling multiple cluster hypotheses (e.g., $N_s=6$) and repeatedly calculating global geometric rewards (intra/inter-cluster distances). The paper lacks a quantitative analysis of the additional time and space complexity introduced by this sampling and evaluation process.

2 The only non-parametric baseline, DeepDPM, is unfairly disadvantaged by the authors' simplistic feature concatenation, which fails to capture cross-view interactions.

3 The experimental design fails to separately validate the "ability to autonomously discover K" from the "performance of the clustering algorithm itself."

4 The paper proposes two modes: single-agent and multi-agent collaboration. But provides no explanation of the threshold for triggering each mode (i.e., when to use a single agent versus when to activate multiple agents).

5 Table 5 is poorly formatted. It lacks explicit definitions for the symbols used, and citation references for the datasets are missing.

---

> ### Author Rebuttal · Authors · 2026-03-30
>
> We sincerely appreciate your recognition of GROK's novelty and our experimental validation. Your insights were invaluable in refining the manuscript.
>
> **Q1.Complexity analysis.**
>
> **A1.** Thank you for your helpful suggestion. Let $N$ denote the number of samples, $d$ the feature dimension, $N_B$ the buffer size, $K_{pred}$ the current predicted $K$, and $T_{\mathcal{F}}$ the state encoding time.
>
> **Time complexity:** The total overhead is $O(N^2 d + NK_{pred} + N_B T_{\mathcal{F}})$. This primarily includes contrastive learning ($O(N^2 d)$), sample-to-centroid assignment ($O(NK_{pred})$), and policy model ($O(N_B T_{\mathcal{F}})$).
>
> **Space complexity:** Memory usage is $O(N^2 + NK_{pred} + N_B M_S)$, driven by the contrastive similarity mask ($O(N^2)$), clustering assignments ($O(NK_{pred})$), and the state $M_S$ in the experience pool ($O(N_B M_S)$).
>
> The overall theoretical complexity is bounded by $O(N^2)$.
>
> **Empirical Comparison:** As shown in follow table, the runtime of GROK is significantly superior to that of DeepDPM.
>
> |Time(s)|Synthetic3d|DHA|CCV|CIFAR10|
> |:-|:-:|:-:|:-:|:-:|
> |DeepDPM|210|163|385|3250|
> |**GROK**|**65**|**116**|**227**|**1054**|
> |**Speedup**|**3.23x**|**1.41x**|**1.70x**|**3.08x**|
>
>
> **Q2.Comparison with DeepDPM.**
>
> **A2.** Good suggestion! To ensure a more fair comparison, we conducted experiments on three datasets by selecting only a single view (View 1). The results, as shown below, demonstrate that GROK outperforms DeepDPM in both clustering performance and $K$ prediction accuracy.
>
> |Dataset| |**Prokaryotic**| | | | |**Hdigit**| | | | |**DHA**| | |
> | :--- | :---: | :---: | :---:|:---:|:---:|:---:|:---:|:---:|:---:|:---:|:---:|:---:|:---:|:---:|
> | |ACC|NMI|$K_{pred}$|$\Delta$||ACC|NMI|$K_{pred}$|$\Delta$| |ACC|NMI|$K_{pred}$|$\Delta$|
> |DeepDPM|0.391|0.029|5| +1||0.288|0.243|6|-4| |0.148|0.302|4|-19|
> |**GROK**|**0.617**|**0.420**|**4**|**0**||**0.567**|**0.523**|**12**|**+2**||**0.478**|**0.607**|**25**|**+2**|
>
> **Q3.Decoupled verification of Auto-K and clustering effectiveness.**
>
> **A3.** Thank you for your insightful suggestion. We have decoupled and validated GROK's autonomous $K$ discovery against its clustering performance using $K_{GT}$.
> | Method | |**Prokaryotic** | | | |**Caltech7** | | | |**CCV** | | | |**CIFAR10** | |
> | :--- | :---: | :---: | :---: | :---: | :---: | :---: | :---: | :---: | :---: | :---: | :---: | :---: | :---: | :---: | :---: |
> | | ACC | NMI | ARI | | ACC | NMI | ARI | | ACC | NMI | ARI | | ACC | NMI | ARI |
> |GROK_GT|0.481|0.171|0.126||0.419|0.538|0.319||0.316|0.305|0.155||0.989|0.974|0.977|
> |**GROK_AutoK**|**0.632**|**0.337**|**0.226**||**0.636**|**0.626**|**0.536**||**0.369**|**0.336**|**0.186**| | **0.993** |**0.981**| **0.985** |
>
> The results show that GROK_AutoK consistently outperforms GROK_GT in clustering metrics. In imbalanced datasets (e.g., Prokaryotic, Caltech7), GROK predicts a lower $K$ than the $K_{GT}$. This suggests the model prioritizes discriminative manifold structures over minority classes with ambiguous boundaries, leading to more robust partitions. Conversely, on CCV and CIFAR10, the model demonstrates strong inherent clustering capabilities, and the agent achieves high prediction accuracy. This confirms that the feedback module successfully transforms predicted $K$ into structural guidance.
>
> **Q4.Multi-agent mode switching threshold.**
>
> **A4.** Thank you.Due to space constraints, please refer to response to Reviewer HhLz (A2).
>
> **Q5.About Table 5.**
>
> **A5.** Thank you.Table 5 summarizes the number of samples ($N$), views ($V$), clusters ($K$), and dimensions (Dims) for eight datasets, with references provided in Section 4.1. In the revised version, Table 5 will be further refined.
>
>
> **Q6.Why choose GRPO over traditional Actor-Critic or PPO algorithms?**
>
> **A6.** Thank you.GRPO was selected primarily for its unique group relative advantage estimation mechanism. Compared to Actor-Critic architectures, it eliminates the need to train an auxiliary Critic network, resulting in a more lightweight and efficient framework. Unlike standard PPO, GRPO utilizes the mean and standard deviation of intra-group samples as a baseline. This approach more effectively filters out reward noise caused by the non-stationarity of feature representations in unsupervised learning, thereby significantly reducing the variance of gradient estimation.
>
> **Q7.About overlapping intervals.**
>
> **A7.** Thank you.The overlapping intervals are designed to eliminate exploration gaps at search space boundaries. By establishing joint search anchors (e.g., $K_{mid}$) between adjacent agents, the system ensures redundant sampling and cross-validation across multiple local policies. This redundancy not only increases boundary sampling density but also provides a unified benchmark for reward alignment. Consequently, it ensures numerical continuity in the global prediction distribution, significantly enhancing decision consistency and preventing exploration collapse.

---

> > ### Author Rebuttal · Reviewer_3x17 · 2026-04-01
> >
> > The authors addressed my concerns.

---

> > > ### Author Response · Authors · 2026-04-05
> > >
> > > We would like to express our sincere gratitude for your positive evaluation and sincerely appreciate the thoroughness and care you put into reviewing our work.

---

### Official Review · Reviewer_KccR · 2026-03-02

**Soundness:** 2
**Presentation:** 3
**Significance:** 3
**Originality:** 3
**Overall Recommendation:** 4
**Confidence:** 3

**Summary:**

The paper proposes GROK, a framework for multi-view clustering without prior knowledge of the number of clusters. It integrates Group Relative Policy Optimization (GRPO) into a perception–decision–feedback loop: a structure-aware backbone for representation learning, a GRPO-based agent to estimate cluster number 𝐾 and a geometric feedback mechanism to refine representations. Experiments on eight datasets show improvements over DeepDPM and competitive results compared to parameterized multi-view clustering baselines.

**Compliance With Llm Reviewing Policy:**

Affirmed.

**Final Justification:**

Thanks for the authors' reply and I am willing to raise my evaluation on this paper.

If the dependency on $K$ can be eliminated without introducing other sensitive hyper-parameters, then this is good for MVC community. It is not yet clear whether the specific settings on each dataset depend on a particular parameter adjustment strategy, so the reproducible code will be useful (as commented by Reviewer vvAa).

One more minor suggestion: The text in Figure 1 is not embedded, and too small text in Figures 2~7, which results in a poor reading experience.

Best regards,

Reviewer KccR

**Key Questions For Authors:**

Plz see weaknesses.

**Limitations:**

Yes.

**Strengths And Weaknesses:**

++ Tackles an important problem: clustering with unknown 𝐾 in multi-view settings. Introduces reinforcement learning (GRPO) into clustering, which is a novel application domain.

++ The perception–decision–feedback loop is conceptually appealing, aiming to couple representation learning with cluster number estimation. Empirical results show strong performance on several benchmarks, particularly in terms of accuracy and robustness to mis-specified 𝐾.

-- The paper claims to be the first to integrate RL into unknown-𝐾 clustering, but prior works (e.g., RGC, ICGR) already explored RL/bandit approaches for cluster number estimation. The novelty lies mainly in swapping in GRPO, which feels incremental rather than groundbreaking.

-- The reward function (Eq. 3) is heuristic and sensitive to high-dimensional variance. No theoretical analysis or convergence guarantees are provided. The divide-and-conquer action space strategy is ad hoc; the necessity and superiority over simpler search methods are not rigorously justified.

-- Comparisons with DeepDPM are unfair: DeepDPM is single-view yet GROK is multi-view. Concatenating features for DeepDPM weakens its performance. No ablation studies are provided to isolate the contributions of GRPO, divide-and-conquer, and geometric feedback. It is unclear which component drives the improvements. Computational cost is ignored. RL-based multi-agent training is likely expensive, but runtime and scalability analyses are missing.

-- On datasets like Prokaryotic and Caltech7, GROK predicts incorrect 𝐾 values but still reports higher ACC. This suggests the method is not reliably estimating 𝐾, undermining its core claim. The emphasis on accuracy improvements overshadows the fact that cluster number estimation sometimes fails significantly.

-- The paper repeatedly emphasizes “first GRPO-based clustering framework,” which risks overstating novelty. The contributions are framed as major breakthroughs, but the work reads more as an engineering combination of existing ideas without deep theoretical insight.

---

> ### Author Rebuttal · Authors · 2026-03-30
>
> We sincerely appreciate your recognition of our work in addressing the important problem of unknown-$K$ MVC, as well as your acknowledgement of the framework's structure and experimental design. We have conducted additional experiments to address your concerns; however, due to space, the detailed results are provided in our responses to other reviewers. We sincerely request the reviewer to provide a more reasonable assessment of this work.
>
> **Q1.Addressing the misconception regarding our novelty claims.**
>
> **A1.** Thank you. We respectfully clarify that we do not claim to be the first to integrate RL into unknown-$K$ clustering, but rather present the first implementation of a GRPO-driven autonomous framework for MVC.While RGC and ICGR (cited in Section 2.2) are tailored for graph-specific clustering, GROK distinguishes itself as the first GRPO-based method designed for general multi-view data. The core innovation of GROK originates from our profound reflection as detailed in the Section 3.1.
>
> This represents a paradigm shift rather than a modular swap. First, GROK replaces absolute evaluation with GRPO-driven relative comparison, mimicking human multi-hypothesis reasoning. It uniquely filters unsupervised variance and resolves the moving-baseline convergence issues typical of traditional RL. Second, unlike prior bandit/RL methods, GROK establishes a Perception-Decision-Feedback loop that back-propagates geometric rewards into manifold reshaping, achieving a deep coupling of representation learning and $K$-estimation.
>
> **Q2.Reward function, divide-and-conquer and comparison with simpler methods.**
>
> **A2.** Great comment! The reward function is rooted in classical manifold metrics (separation and compactness), not mere heuristics. While high-dimensional variance is universal, GROK leverages GRPO’s group-relative mechanism to resolve this at a mechanistic level. By calculating relative advantages within a group, consistent background noise across concurrent samples is mathematically canceled. This internal baseline subtraction ensures stable relative rankings even in high-dimensional spaces.
>
> Divide-and-conquer: Section 4.4 (Fig. 4) shows that multi-agent  is a logical necessity for maintaining stability and accuracy in high-cardinality scenarios. While prior works like RGC are restricted to a limited search space ($K \leq 10$), GROK addresses broader exploration ($K$ up to 40). In high action spaces, a single-agent search is prone to exploration collapse. Our strategy decomposes this complexity, enabling robust convergence that single-agent RL cannot achieve.
>
> GROK achieves higher accuracy in $K_{pred}$ than DeepDPM, a leading non-parametric method, demonstrating its effectiveness.
>
> **Q3.Comparisons with DeepDPM, ablation studies and runtime analyses.**
>
> **A3.** Comparison with DeepDPM: Please refer to Reviewer 3x17 (A1), as space here is limited.
>
> Ablation studies: We emphasize that the necessity of our core modules is validated through ablation studies (Sec. 4.3-4.4). About the divide-and-conquer—a point also addressed in A2. About the geometric feedback, results in Tab.3 prove it leads to a significant performance boost in ACC. It confirms that our feedback-driven feature reshaping creates a unique synergy between representation learning and $K$-estimation—a critical integration absent in prior works.
>
> Runtime analyses: Unlike the the assumption that multi-agent is inherently expensive, single-agent search suffers from slow convergence in high-cardinality spaces. GROK's multi-agent accelerates the process by decomposing global searches into parallel, lightweight sub-tasks. Empirical results confirm GROK’s efficiency. Please refer to Reviewer 3x17 (A2), as space here is limited.
>
> **Q4.Analysis of results on Prokaryotic and Caltech7.**
>
> **A4.** We respectfully disagree that $K$-deviations on Prokaryotic and Caltech7 undermine our claims; rather, they demonstrate GROK’s robustness in capturing intrinsic structure. As already detailed (Line 270), these datasets feature class imbalance and low-discriminability tail clusters. RGC notes that clustering peaks near $K_{GT}$ often reveal more intrinsic data distributions. Due to space, please refer to Reviewer 3x17 (A3) for our experiments comparing GROK using $K_{GT}$ versus $K_{pred}$.
>
> **Q5.Clarification of the "First GRPO-based clustering framework" statement.**
>
> **A5.** We would like to tone down this statement in the revised version. We would appreciate if you can direct us to existing literature that utilizes GRPO to solve the unknown-$K$ MVC problem. We would be more than happy to cite and compare them in the revised version.
>
> Reviewers vvAa, HhLz, and 3x17 recognized our work’s novelty, with vvAa specifically labeling it groundbreaking. We clarify that we never claimed to be the "first to introduce RL into clustering." As the concern over "overstated innovation" likely stems from a misunderstanding, we respectfully request a reassessment of GROK.

---

> > ### Author Rebuttal · Reviewer_KccR · 2026-04-03
> >
> > Thanks for the responses that addressed most of my concerns. Nevertheless, I also agree with Reviewer vvAa that this algorithm introduces additional hyperparameters. If the aim is to address the need for the parameter K, but other more sensitive parameters are introduced instead. This seems not to fundamentally solve the problem.

---

> > > ### Author Response · Authors · 2026-04-05
> > >
> > > We are very pleased to learn that our responses addressed most of your concerns. Regarding the concern about additional hyperparameters, we would like to offer further clarification to demonstrate that our approach offers a fundamental advancement in learning the cluster number $K$.
> > >
> > > **1. Generalizability of Hyperparameters:** As detailed in our response to Reviewer vvAa (A6), with $K_{max}$ serving as a generous search boundary, all datasets in our experiments share an identical set of hyperparameters. This consistency underscores that our model is not sensitive to parameter tuning and possesses strong robustness across diverse data distributions.
> > >
> > > **2. Significant Progress over Prior Arts:** You mentioned the baseline methods RGC and ICGR. In RGC, the true $K$ ranges from 4 to 8, and $K_{max}$ is uniformly set to a small value of 10. In contrast, our work tackles much more challenging scenarios where the true $K$ ranges from 3 to 31. By setting $K_{max}$ between 10 and 40, our method infers the exact ground-truth $K$ or a highly accurate approximation on Prokaryotic and Caltech7 datasets, which effectively captures the underlying semantic structures. This capability to operate within a significantly larger search space represents a substantial leap forward compared to previous multi-armed bandit-based approaches.
> > >
> > > **3. The Necessity of the RL Framework:** The transition from simple multi-armed bandits to our proposed GRPO framework is a strategic evolution to address the unknown $K$ problem in clustering. While $K_{max}$ acts as a search boundary, it is far less restrictive than requiring an exact $K$. Our framework demonstrates that this is a robust and scalable trajectory toward achieving fully autonomous clustering.
> > >
> > > While we appreciate your critical perspective, we hope our clarifications for Q1 and Q5 have provided a satisfactory response to your concerns. Given that the other reviewers have shared a positive outlook on the novelty of this work, we kindly request a reconsideration of our manuscript’s overall value and contributions.
> > >
> > > Finally, we would like to express our sincere gratitude once again for the time and effort you have dedicated to reviewing our work. Please do not hesitate to contact us if you have any further questions.

---

### Official Review · Reviewer_vvAa · 2026-03-11

**Soundness:** 3
**Presentation:** 3
**Significance:** 3
**Originality:** 3
**Overall Recommendation:** 5
**Confidence:** 4

**Summary:**

This paper proposes GROK, a novel framework for multi-view clustering with an unknown number of clusters. The authors pioneer the adaptation of Group Relative Policy Optimization (GRPO), originally developed for Large Language Model reasoning, to the domain of unsupervised learning. GROK establishes a "Perception-Decision-Feedback" closed-loop system driven by a clustering decision agent.

**Compliance With Llm Reviewing Policy:**

Affirmed.

**Final Justification:**

During the rebuttal, authors effectively addressed my concerns. I believe this is a well-researched and experimentally rich article.

**Key Questions For Authors:**

- How is the cluster centroid set $C$ obtained?
- What is the complexity of the method? The authors should include a detailed analysis of training time, FLOPs and efficiency comparisons with baselines (e.g., DeepDPM).
- Please provide more detailed information regarding the hyperparameters. Specifically, clarify whether a single set of hyperparameters was shared across all datasets or if they were fine-tuned individually for each dataset.

**Limitations:**

yes

**Strengths And Weaknesses:**

Strength：
- The proposed GRPO-based custer decision agent strategy is a groundbreaking attempt to utilize reinforcement learning strategies  to solve the discrete search problem of cluster numbers.
- The proposed "Perception-Decision-Feedback" loop effectively addresses the disconnection between feature learning and
$K$ estimation in traditional methods.

Weakness:
- The proposed method is incremental，requiring three components and two stages of training. In order to adaptively determine hyperparameter $K$, authers introduce a large number of parameters and prior knowledge.
- While the paper mentions using a lightweight policy network, introducing Reinforcement Learning (especially multi-agent collaboration and group sampling) typically increases training time and computational overhead.
- The paper lacks implement details, which raises serious concerns about its reproducibility. Specifically, the architectural configurations of the three core components—the multi-view autoencoder backbone, the Transformer-based fusion module, and the Scalable Policy Framework—are not provided. Without these structural details, it is impossible to verify whether the comparisons with baseline methods are fair and equitable.

---

> ### Author Rebuttal · Authors · 2026-03-30
>
> We sincerely appreciate your recognition of our work as a groundbreaking adaptation of GRPO and your positive feedback on our Perception-Decision-Feedback loop. We are also deeply grateful for your thorough review and valuable suggestions to improve our work. We have conducted additional experiments to address your concerns; due to space, the results are provided in our responses to other reviewers.
>
> **Q1.Clarification of the proposed framework and hyperparameters.**
>
> **A1.** Thank you. We would like to clarify that while many SOTA methods employ a two-stage training process (pre-training for feature representation followed by fine-tuning for cluster-friendly features), this common paradigm does not imply that our work is an incremental improvement. The core innovation of GROK lies in the construction of an organic closed-loop system integrating Perception, Decision, and Feedback. Unlike existing $K$-prediction methods that typically perform unidirectional inference based on static features, GROK introduces a dynamic feedback mechanism. Specifically, $K_{pred}$ is fed back into the Perception module in real-time to refine state observations, which subsequently enhances the accuracy of the decision module. The interdependence of these components is essential for achieving automated $K$ prediction while maintaining stable clustering performance.
>
> Hyperparameters: Please refer to A6 for the analysis of parameter counts. In essence, the decision module of GROK primarily relies on two hyperparameters: $K_{max}$ and $N_B$ (shared across all datasets).
>
> **Q2. Computational overhead evaluation of GRPO and the multi-agent.**
>
> **A2.** Thank you. In fact, GRPO optimizes GROK's efficiency. Unlike traditional Actor-Critic, GRPO’s design eliminates the need for a heavy value function network, which reduces the model parameters and the computational overhead. Furthermore, multi-agent is engineered to enhance search efficiency rather than adding extra burden. In high-cardinality action spaces, a single agent often struggles with sparse reward signals and slow convergence. By decomposing the global search into localized sub-tasks, each policy network operates within a restricted action interval. Crucially, these agents are executed in parallel.
>
> We have added the runtime analysis of GROK and DeepDPM in Reviewer 3x17 (A1). The results show GROK's superior efficiency.
>
> **Q3. Reproducibility.**
>
> **A3.** Thank you. Detailed network architectures are provided in Appendix A. The source code will be made publicly available immediately upon acceptance.
>
> **Q4.Clustering centroids.**
>
> **A4.** Great comment! $\mathcal{C}$ is obtained using K-means after the agent fixes the current $K_{pred}$. Specifically, the consensus features $U$ are partitioned based on $K_{pred}$ to generate $\mathcal{C}=\{c_{1},...,c_{K_{pred}}\}$. We will add this to the revised version.
>
> **Q5. Complexity and comparison with DeepDPM.**
>
> **A5.** Thank you for your valuable suggestions. Let $N$ denote the number of samples, $d$ the feature dimension, $N_B$ the experience buffer size, $K_{pred}$ the current predicted clusters, and $T_{\mathcal{F}}$ the state encoding time.
>
> Time complexity: The total overhead is $O(N^2d+NK_{pred}+N_BT_{\mathcal{F}})$. This primarily includes contrastive learning ($O(N^2 d)$), sample-to-centroid assignment ($O(NK_{pred})$), and policy model  ($O(N_BT_{\mathcal{F}})$).
>
> Space complexity: Memory usage is $O(N^2+NK_{pred}+N_BM_S)$, driven by the contrastive similarity mask ($O(N^2)$), clustering assignments ($O(NK_{pred})$), and the state $M_S$ in the experience pool ($O(N_BM_S)$).
>
> FLOPs: The computational load is approximately $O(N^2d+MINK_{pred}d)$, where $M$ and $I$ represent the number of RL evaluations and K-means iterations, respectively.
>
> The overall theoretical complexity is bounded by  $O(N^2)$.
>
> Comparison with DeepDPM: Please refer to Reviewer 3x17 (A1) for a detailed discussion, as space is limited.
>
> **Q6.Hyperparameter.**
>
> **A6.** Thank you. Backbone hyperparameters are kept consistent with baselines for fair comparison. The hyperparameter of GROK primarily focuses on the policy model, which can be categorized into two groups:(1)Native GRPO hyperparameters: these follow the standard settings of GRPO to ensure training stability, including the group sample size $N_s = 6$, the number of policy network iterations (20), and the clipping range $\epsilon = 0.2$.
> (2)Clustering-specific hyperparameters: To adapt GRPO to the clustering, we introduced targeted configurations, including the experience buffer size $N_B = 30$  and the search space upper bound $K_{max} \in [10, 40]$.
>
> It is important to emphasize that, with the exception of $K_{max}$, all datasets share an identical set of hyperparameters. The $K_{max}$ for each dataset are as follows:
>
> |Dataset|$K_{GT}$|$K_{max}$|
> |:-:|:-:|:-:|
> |Prokaryotic|4|10|
> |Synthetic3d|3|10|
> |Caltech7|7|20|
> |Hdigit|10|20|
> |CIFAR10|10|20|
> |CCV|20|40|
> |DHA|23|40|
> |YouTubeFace|31|40|

---

> > ### Author Rebuttal · Reviewer_vvAa · 2026-04-03
> >
> > Thanks for the responses. Most of my concerns have been addressed.

---

> > > ### Author Response · Authors · 2026-04-05
> > >
> > > We are deeply grateful to the reviewer for the positive recognition and the increased score. We are very pleased that our revisions have addressed your concerns and improved the quality of the work.

---

### Official Review · Reviewer_HhLz · 2026-03-12

**Soundness:** 4
**Presentation:** 3
**Significance:** 4
**Originality:** 4
**Overall Recommendation:** 5
**Confidence:** 5

**Summary:**

This paper proposes a novel end-to-end framework named GROK, designing to address the challenge of applying contrastive multi-view clustering with prior knowledge lacking and the unknown cluster number $K$. The framework pioneers the introduction of Group Relative Policy Optimization (GRPO) into the multi-view clustering domain to construct a Perception-Decision-Feedback closed-loop system. GRPO could be viewed as a reinforcement learning strategy originally used for LLM reasoning. In the perception phase, a structure-aware adaptive backbone is employed to extract consistent and discriminative consensus representations ; in the decision phase, a cluster decision agent executes a divide-and-conquer strategy, performing group sampling within a discrete action space of candidate $K$ values and autonomously determining the optimal $K$ via an adaptive reward function; and in the feedback phase, a geometric clustering guidance mechanism transforms the agent's $K$ prediction into explicit differentiable constraints to optimize feature manifolds and reshape the latent space. Experimental results across eight datasets demonstrate that GROK not only accurately estimates the cluster number but also achieves SOTA-level clustering performance while possessing the capability to enhance existing models as a plug-and-play module.

**Compliance With Llm Reviewing Policy:**

Affirmed.

**Key Questions For Authors:**

Please see weakness

**Limitations:**

Yes.

**Strengths And Weaknesses:**

Strengths：
1.	The manuscript exhibits a well-defined motivation, clear descriptions, a well-organized structure, and comprehensive experimental validation.
2.	This work presents the first integration of GRPO into multi-view clustering to ingeniously resolve the heavy reliance of existing methods on a pre-defined cluster number $K$. GRPO attempts to handle an advanced reinforcement learning strategy originally designed for LLM reasoning, which is interesting and valuable.
3.	The proposed multi-agent collaborative divide-and-conquer strategy successfully addresses the challenges of high-cardinality action spaces, with its effectiveness rigorously demonstrated through experimental results.
4.	Experimental results verify that GROK can be integrated as a universal plug-and-play module into existing frameworks (e.g., MFLVC, GCFAggMVC, and SEM), significantly enhancing their performance without requiring manual $K$ configuration.

Weaknesses:
1.	Please explicitly state which specific algorithm (e.g., K-means.) is employed by the Cluster Structure Solver in Section 3.4 to obtain the pseudo-labels $\hat{Y}$.
2.	The paper proposes a divide-and-conquer strategy to handle high-cardinality action spaces. However, the descriptions of the partitioning thresholds (e.g., $K_{mid}$) and the critical point for transitioning from single-agent to multi-agent mode are not sufficiently detailed. Are these interval sizes automatically calculated through equal partitioning based on $K_{max}$, or are they determined by a set of predefined empirical values?
3.	To balance sensitivity across varying cluster numbers, a dual-scale normalization is adopted in Equation (3). Does a discontinuity in reward values occur at the transition point from the linear scale $(\psi \propto K)$ to the logarithmic scale $(\psi \propto \log K)$? Furthermore, would such a potential discontinuity affect the convergence stability of the agent?
4.	The manuscript currently lacks an intuitive architectural flowchart or formal pseudocode. It is recommended to include an algorithm table that clearly delineates the sequential order of the group sampling phase, the environmental feedback phase, and the policy update phase to improve the overall clarity of the framework.

---

> ### Author Rebuttal · Authors · 2026-03-30
>
> We sincerely appreciate the reviewer's positive assessment of our work’s novelty, experimental validation of GROK. We are also grateful for your constructive suggestions, which have significantly helped improve the quality of our manuscript.
>
> **Q1.Explicit identification of the cluster structure solver in Section 3.4.**
>
> **A1.** Thank you for this helpful suggestion.  The cluster structure solver in Section 3.4 employs the K-Means algorithm. Specifically, it uses the predicted cluster number $K_{pred}$ finalized by the cluster decision agent to partition the consensus representation $U$, thereby generating the centroid set $\mathcal{C}$. We have added Section 3.4 in the revised manuscript to explicitly state this implementation detail.
>
> **Q2.Detailed logic of the divide-and-conquer partitioning thresholds.**
>
> **A2.** Thank you for the insightful question. The interval sizes are automatically calculated based on $K_{max}$ using equal partitioning with a fixed interval range of 20. Specifically, a single agent handles the action space $[2, 20]$. Once $K_{max} > 20$, the multi-agent collaborative mode is automatically activated. This design ensures sufficient sampling density across the entire search range, preventing exploration collapse in high-cardinality scenarios. Our choice of the 20-unit interval stems from observations of standard multi-view clustering datasets, where cluster numbers typically range from 3 to 100, with the most common benchmarks falling within 5–20. By testing eight datasets with $K$ ranging from 3 to 31, we rigorously verified that this strategy adapts seamlessly to varying cluster cardinalities. We have added these details to Section 3.3.2 of the revised manuscript.
>
> **Q3.Reward value continuity during dual-scale normalization and its impact on convergence stability.**
>
> **A3.** Thank you. We address potential reward discontinuity through two synergistic mechanisms: 1) Segmented governance with overlapping intervals, where the multi-agent mode confines policy updates to localized ranges while using $K_{mid}$ as a joint anchor for redundant sampling and cross-verification. This creates a unified reward alignment baseline, ensuring a smooth transition of the global action distribution across boundaries. 2) Relative optimization via GRPO, which utilizes group relative advantage ($A_i$) to shift the focus from absolute reward values to relative performance. By filtering out numerical fluctuations at transition points, this approach inherently maintains robust convergence stability and logical consistency across the entire search space.
>
> **Q4.Request for a formal pseudocode to improve framework clarity.**
>
> **A4.** We appreciate this constructive suggestion to improve the readability of our work. In the revised version, we have included a formal algorithm table in the Section 3. The sequential workflow is summarized as follows:
>
> **Phase I: Structure-aware pre-training.**
>     We initialize the backbone network $\Phi$ using contrastive learning to construct a discriminative consensus representation $U$. Features from multi-views are fused via a Transformer-based attention mechanism and optimized using reconstruction MSE and structure-guided contrastive loss to preserve the data's inherent manifold topology (Eqs. 1–2).
>
> **Phase II: Joint optimization & fine-tuning.**
>     This phase follows a Perception-Decision-Feedback closed-loop logic over 50–100 epochs, utilizing a partitioned action space to manage diverse cluster counts.
>
> **1.Perception & state construction:** The backbone $\Phi$ extracts features $U$, which are partitioned via K-Means to obtain centroids $\mathcal{C}$. The RL state $s_t$ is constructed by fusing global feature means with $\mathcal{C}$ to represent both global distribution and local structural density.
>
> **2.Multi-policy group decision:** To efficiently search the action space $K \in [2, K_{max}]$, the space is partitioned among specialized policy networks $\pi_\theta$ (e.g., Policy 1 for $K \in [2, 20]$, Policy 2 for  $K \in [20, 40]$). Each active policy performs group sampling via temperature sampling based on $s_t$, and the environment calculates a geometric reward $R$ for each sampled $K$ (Eq. 3).
>
> **3.Policy update and optimization:** The agent computes group relative advantages ($A_i$ in Eq. 4) using samples within each group. Each partitioned $\pi_\theta$ is then independently optimized for $N_{iter}$ iterations using the GRPO objective function (Eq. 5), effectively locking the optimal prediction $K_{pred}$ by maximizing the advantage of superior hypotheses.
>
> **4.Geometric feedback:** The $K_{pred}$ is selected and transformed into a geometric clustering constraint (Eq. 6) to reshape the feature manifold for the next iteration.
>
> This iterative mechanism ensures that representation learning and cluster number estimation are deeply coupled and mutually optimized.

---

> > ### Author Rebuttal · Reviewer_HhLz · 2026-04-01
> >
> > The questions I raised have been fully resolved.

---

> > > ### Author Response · Authors · 2026-04-05
> > >
> > > We are sincerely grateful for your recognition of our work and truly appreciate the time and effort you invested.

---

### Decision · Program_Chairs · 2026-04-30

**Decision:**

Accept (regular)

**Comment:**

This paper proposed the GROK framework, and reviewers generally found the overall idea technically sound and  well motivated.
The rebuttal successfully addressed most of the major concerns, several reviewers explicitly stated that their concerns were fully resolved, and one reviewer further raised their evaluation after the follow-up discussion.
Overall, I lean toward acceptance.